# Selective Copper Recovery by Acid Leaching from Printed Circuit Board Waste Sludge

**Ha Bich Trinh, Seunghyun Kim and Jaeryeong Lee *** 

Department of Energy and Resources Engineering, Kangwon National University, Chuncheon, Gangwon-do 24341, Korea; hab.trinh@gmail.com (H.B.T.); rlatmdgus930@kangwon.ac.kr (S.K.)
* Correspondence: jr-lee@kangwon.ac.kr; Tel.: +82-33-250-6252

**Abstract:** The most challenging issue associated with recycling the sludge generated from printed circuit boards (PCBs) is the separation of copper (Cu) from iron (Fe), using multi-stage leaching, or adding oxidizing and precipitating agents. Herein we investigated simple acid leaching to effectively extract copper and limit iron dissolution. Selective copper leaching was achieved with all the acids studied, including HCl, $HNO_3$, and $H_2SO_4$. The lower concentration of acid solutions resulted in a larger difference in leachabilities between Cu and Fe. Among three leachates, the $H_2SO_4$ solution performed effectively on the selective leaching of Cu and Fe. Adjusting the pulp density to 4% and the $H_2SO_4$ concentration at ~0.2 M, accomplished ~95% Cu leaching and reduced the Fe extraction to less than 5%. Kinetic studies revealed that Cu leaching followed the ash diffusion-controlled mechanism. Aactivation energy ($E_a$) of 9.8 kJ/mol was determined for the first 10 min of leaching. Further, leaching up to 60 min corresponded to a mixed control model, increasing the $E_a$ to 20.9 kJ/mol. The change in the control model with regard to the two leaching stages can be attributed to the Cu hydroxide and metal phases present in the original sample. A simple, economically attractive $H_2SO_4$ acid leaching process was demonstrated, recovering Cu efficiently and selectively from PCBs waste sludge under moderate conditions.

**Keywords:** copper; selective recovery; acid leaching; PCBs waste sludge

## 1. Introduction

The high demand for electronic products in modern life has rapidly expanded the production of printed circuit boards (PCBs), resulting in the release of a massive amount of unwanted solid and liquid wastes containing hazardous materials [1]. Manufacturing PCBs is a complicated multi-stage process, using a wide range of valuable metals, such as Au, Cu, Ni or Sn. Therefore, these metal-bearing wastes have to be managed in terms of environmental protection and economic benefits [2]. The recycling of such secondary resources will reduce the negative effect of harmful wastes and prevent the depletion of the primary metal ores, which is important for the development of a sustainable metallurgical industry. The compositions of wastes generated from PCBs manufacturing depend on the specific stage or the chemicals used during processing [3]. The wastes can be in solid or liquid form, containing organic or inorganic compounds. This requires proper treatments and recycling methods before the final disposal into the environment. Recently, waste sludge from PCBs manufacturing has been considered as a resource with high copper recycling potential [2]. Several methods have been employed to recover copper from PCBs waste sludge, primarily based on their chemical compositions, and conventional approaches include pyrometallurgical or hydrometallurgical methods [4].

In the pyrometallurgical route, the sludge was mixed with suitable fluxes and separated from iron during smelting. Further purification is necessary, such as fire- and electro-refining, to enhance the quality of copper. Moreover, it should consider several disadvantages, such as a large amount of

energy consumption, harmful by-products generation, and loss of copper to the slag [4]. Therefore, acid leaching has been effectively employed as an alternative process to recover copper from PCBs waste sludge, following the hydrometallurgical route. A wide range of acids was investigated to extract copper; for example, inorganic acids like HCl, $HClO_4$, $HNO_3$, and $H_2SO_4$, organic acids like citric acid and acetic acid, and spent acidic PCBs etching solutions [5–9]. Acid leaching is a simple process that extracts target metals from the sludge sample in a short duration with certain high efficiency and can combine with further separation stages such as precipitation, adsorption, or solvent extraction [8]. Investigations on sulfuric acid leaching of PCBs production sludge using a full factorial experiments design revealed the significant influence of sulfuric acid concentration, pulp density and leaching time; and there was 0.4% difference between predict value (97%) and experimental extraction (96.8%) at optimal conditions ($H_2SO_4$ 0.84 M, pulp density 1%, time 80 min) [8]. Two-stage leaching and evaporation were applied to extract and purify copper from copper-contaminated sludge: (i) in the first stage, 2.0 N sulfuric acid could leach out 95.2% Cu and other impurities (Fe, Al, and Pb); (ii) in the second stage, an ammonia solution was used to stabilize Cu by forming an amine complex in solution, while other impurities were precipitated as hydroxides; (iii) the final evaporation stage eliminated the ammonia and converted the copper amine complex into copper oxide with a purity of 98.4% [10]. A ferrite process was used as pre-treatment for copper-contaminated sludge, using air as an oxidizing agent to form ferrite complexes. This pre-treatment promoted Cu leaching to 98.29% in the following stage, using a 0.5 N $H_2SO_4$ solution, and reduced the iron dissolution to less than 0.73% [11]. Sulfuric acid leaching (1.0 M, 50 °C for 60 min), followed by a chemical exchange using Fe powder (ratio Fe:Cu 5:1, pH 2.0, 50 °C, 200 rpm), could precipitate 95% copper from the leach liquor [12]. Another leaching process used a spent acid etching solution with the assistance of ultrasound at 300 W and further adjustment of pH by lime to 2.5, obtaining a high Cu leaching efficiency of 93.76% and a noticeably low Fe extraction of 2.07% [9]. Sulfuric leaching followed by the Jarosite process could extract 93% of Cu and remove Fe as the phase of $KFe_3(SO_4)_2(OH)_6$, and later recover Cu using sulfide precipitation, which is suitable for application of pyro-metallurgy process [13]. An alternative approach by bioleaching using thermophilic sulfur-oxidizing bacteria could obtain around 65% Cu after four cycles (40 days) in a sequencing batch reactor [14]. The most challenging issue for hydrometallurgical processes is the separation of copper from iron, which involves several leaching stages, applying ultrasound, using oxidants, or precipitating agents. Consequently, the multiple stages of leaching, separation, and purification consume more chemicals and produce significant amounts of solid and liquid waste. Therefore, it is important to investigate effective processes that can decrease the number of operating stages and separate selectively copper from the ion.

In this study, a simple leaching process using various acids was investigated to selectively recover copper and reduce iron extraction. The type and concentration of acid and the ratio between solid sample and solution can be varied to maximize the leachability of copper and minimize the dissolution of iron from leach liquors. High copper extraction efficiency and low iron selectivity were obtained with one leaching stage. This was achieved without using any other oxidizing or precipitating agents, which is the most significant advantage of the presented work. Further, kinetic studies were applied to study the mechanism of copper dissolution at selected leaching conditions.

## 2. Materials and Methods

The PCBs production process can generate a large amount of wasted water and spent solution, which is commonly neutralized by using alkaline followed by coagulation and sedimentation to produce a metal-containing sludge before sending to final treatment or disposal [8]. The waste sludge in this study was supplied by SungEel Himetal, South Korea, and dried at 105 °C for 24 h. The dried sample was subsequently scrubbed at a speed of 83 rpm for 5 min, using a rod mill with a diameter of 10 mm and a length of 89.74 mm, and sieved to collect the fraction of particle size 45 μm [15]. Wet methods were used to analyze the chemical compositions, and the results are listed in Table 1. Hydrochloric acid (HCl, 35%, Junsei Chemical Co. Ltd., Tokyo, Japan), nitric acid ($HNO_3$, 61%, Junsei

Chemical Co. Ltd., Tokyo, Japan) and sulfuric acid ($H_2SO_4$, 95%, Junsei Chemical Co. Ltd., Tokyo, Japan) were used as lixiviants to leach the copper sludge.

Leaching experiments were performed in a 1000-mL glass beaker at a fixed stirring speed of 250 rpm using a magnetic stirrer bar. A specific amount of ground sample was dissolved in the acidic lixiviant to maintain at desired conditions of the investigated experiment, and filtrated to obtain the leach liquor for the analysis of metal contents. The metal content was analyzed with an inductively coupled plasma spectrometer (ICP, OPTIMA 7300DV, Perkin Elmer, Seoul, South Korea). The ICP results were used to estimate the metal leaching efficiency as:

$$\% \text{ Leaching} = \left( \frac{M_S - M_L}{M_S} \right) \times 100 \tag{1}$$

where $M_S$ and $M_L$ are the metal masses in the initial feed sample and the leach liquor, respectively.

**Table 1.** Chemical compositions of the copper sludge sample.

| Metals | UOM | Composition |
|--------|-----|-------------|
| Cu | | 18.34 |
| Fe | | 29.35 |
| Ca | % | 6.24 |
| Pb | | 2.06 |
| Sn | | 1.72 |
| Mg | | 7134 |
| Al | | 2277 |
| Zn | ppm | 1165 |
| Na | | 1053 |
| Si | | 0 |

UOM: unit of measurement.

## 3. Results and Discussion

### 3.1. Characteristics of Copper Sludge

The sample pH was 7.87, which can be attributed to the prior neutralization using basic agents. Thus, the metals in the sludge sample were presented as hydroxides. The solid sample was analyzed by X-ray diffraction (XRD, D2 Phaser, Bruker, Seoul, South Korea); however, there were no significant Cu or Fe peaks detected, which implies the presence of non-crystalline phases of Cu and Fe in the sludge (Figure 1). The thermal degradation of the sample was investigated with thermogravimetric and differential thermal analyses (TGA-DTA, DTG 60H, Shimadzu, Seoul, South Korea). The sample was heated from 25 °C to 500 °C at a heating rate 20 °C/min, using nitrogen gas at a flow of 40 mL/min. As reported previously, the hydroxide form of Cu or Fe can be identified by dehydration within this temperature range [16–18]. The TGA curve in Figure 2 displays two major mass loss stages, which corresponds to two different DTA peaks. The first stage between 140 °C and 160 °C shows a ~5% weight loss, characterized by an endothermic DTA peak. This is attributed to the decomposition of $Cu(OH)_2$, forming CuO [18,19], following Equation (2). The dehydration of $Fe(OH)_3$ forming FeOOH took place at temperatures below 350 °C, showing an exothermic peak. The complete decomposition of $Fe(OH)_3$ occurred at temperatures above 800 °C [16,20]. $Fe(OH)_2$ was dehydrated to FeO above 375 °C, corresponding to an endothermic peak [17]. Therefore, the second mass loss between 160 °C and 300 °C could involve the dehydration of $Fe(OH)_3$ to FeOOH and that of $Fe(OH)_2$ to FeO. The significantly higher amount of Fe(III) compared to Fe(II) resulted in a large exothermic DTA peak. The sample was heated at 500 °C and was further analyzed by XRD. The results show the appearance of Cu as both metal and oxide (Cu and CuO), whereas Fe was present as Fe (III) oxide hydroxide FeOOH (Figure 1). The TGA-DTA and XRD results certainly confirmed the presence of CuO and FeOOH, due to the decomposition of $Cu(OH)_2$ and $Fe(OH)_3$ precursors in the sample (Equations (2) and (3)).

$$Cu(OH)_2 \rightarrow CuO + H_2O \tag{2}$$

$$Fe(OH)_3 \rightarrow FeOOH + H_2O \tag{3}$$

Although Fe(II) phases could not be identified, Fe(II) compounds most probably existed as amorphous phases, or their contents were lower than the detection limit of the analysis. The possibility of Fe(II) phases should be considered, because PCBs manufacturing is a complex process, which can generate both Fe(II) and Fe(III) compounds, such as $Fe_3O_4$ [1] or hydroxides such as $Fe(OH)_3$ and $Fe(OH)_2$ [4]. It can be concluded that the main constituents in the sludge sample are Cu, such as $Cu(OH)_2$ and Cu metal, whereas Fe exists mainly as $Fe(OH)_3$ and minor amounts of $Fe(OH)_2$. The SEM image in Figure 3 clearly demonstrates the porous morphology on the surface of the sludge sample, due to the presence of metal hydroxides, and the mapping results show major Cu and Fe and minor Ca peaks (Scanning Electron Microscope-Energy Dispersive Spectrometer (SEM-EDS) Jeol-JSM-6380LA, Jeol Korea Ltd., Seoul, South Korea). The chemical analysis identified Cu (18.34%) and Fe (29.35%) as main constituents, while other metals such as Ca (6.24%), Pb (2.06%), and Sn (1.72%) were presented in lower concentrations.

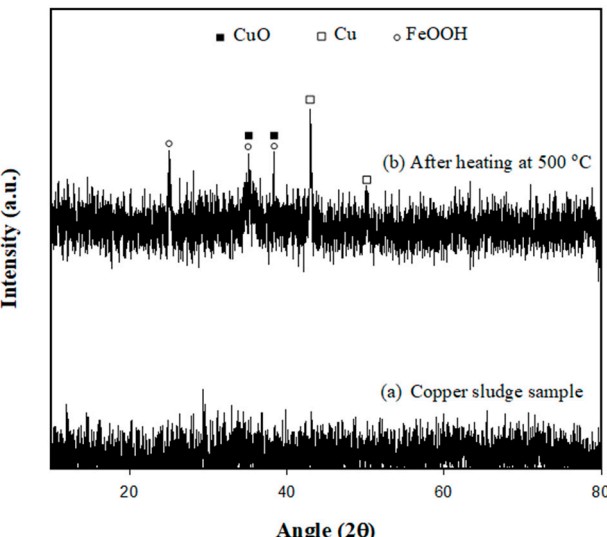

**Figure 1.** X-ray diffraction of the sludge sample before and after heating at 500 °C.

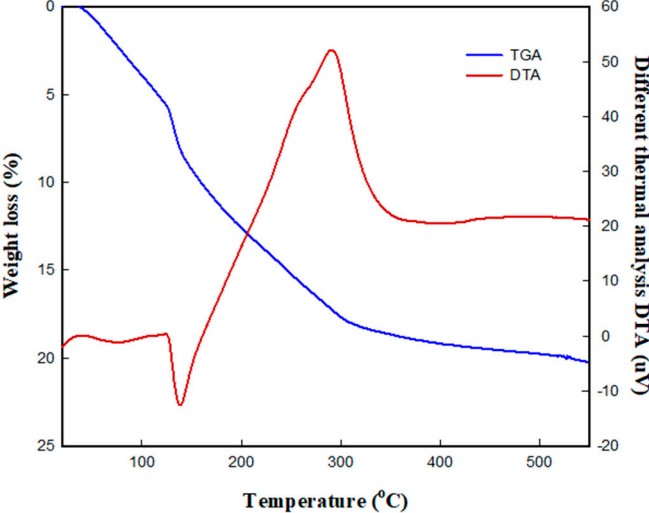

**Figure 2.** DTA-TGA curves of copper sludge sample.

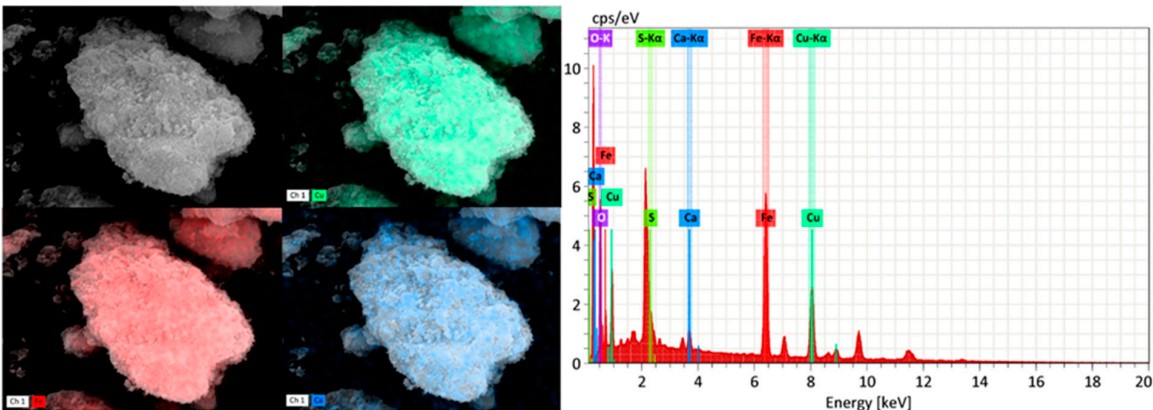

**Figure 3.** Morphology and SEM-EDS of the copper sludge sample.

*3.2. Potential of Selective Copper Leaching from Sludge using Different Acids*

The acid selection can result in different behaviors of metals leaching from the sludge, and mineral acids, such as HCl, HNO$_3$, and H$_2$SO$_4$, are generally more efficient than organic acids [5]. Moreover, Cu(OH)$_2$ and Fe(OH)$_3$ have significantly different solubilities, depending on pH, e.g., Cu(OH)$_2$ has a K$_{sp}$ of $2 \times 10^{-19}$ and Fe(OH)$_3$ shows a K$_{sp}$ of $6 \times 10^{-38}$ [21]. Therefore, the copper sludge sample (5.0 g) was dissolved in 500 mL of acid (HCl, HNO$_3$ and H$_2$SO$_4$), varying the concentration from 0.05 to 0.5 M and maintaining other conditions constant at 25 °C, for 60 min. The results in Figure 4 show that there were significant differences between the acids leaching behavior of Cu and Fe. The Cu dissolution was already high (>70%) at low acid concentrations (0.05 M) and increased in the order of HCl < HNO$_3$ < H$_2$SO$_4$. Small amounts of Fe were leached out at the same conditions, for example ~0.2%, using HCl and HNO$_3$, and 8.6%, using H$_2$SO$_4$ (Figure 4a,b). This leaching results were promising with regards to the separation of Cu from Fe using acids, specifically H$_2$SO$_4$, due to the higher leachability of Cu and the moderate dissolution of Fe. The increase in acid concentration from 0.05 to 0.5 M enhanced the leachability of Cu and Fe; however, the level of Fe leaching was considerably higher than that of Cu. The Cu leaching efficiency only increased 26% in HNO$_3$ and 9% in HCl, varying the concentration from 0.05 M to 0.1 M, and maintained constant up to 0.5 M. The enhancement of Cu leachability in H$_2$SO$_4$ solutions was not significant, but increasing the H$_2$SO$_4$ concentration from 0.05 M to 0.25 M achieved rapid improvement of Fe leaching, from 8.6% up to ~90% (Figure 4c). The same large increase in Fe extraction was obtained in HCl and HNO$_3$ solution. At high acid concentrations of 0.5 M, both Cu and Fe were completely dissolved, which is not preferred. Therefore, the effect of H$_2$SO$_4$ acid concentration was further investigated to obtain high Cu extraction and low Fe leaching. This is the most important point compared to previous studies, because only one single stage and no oxidizing agents are used in this case to extract copper from the copper-containing sludge.

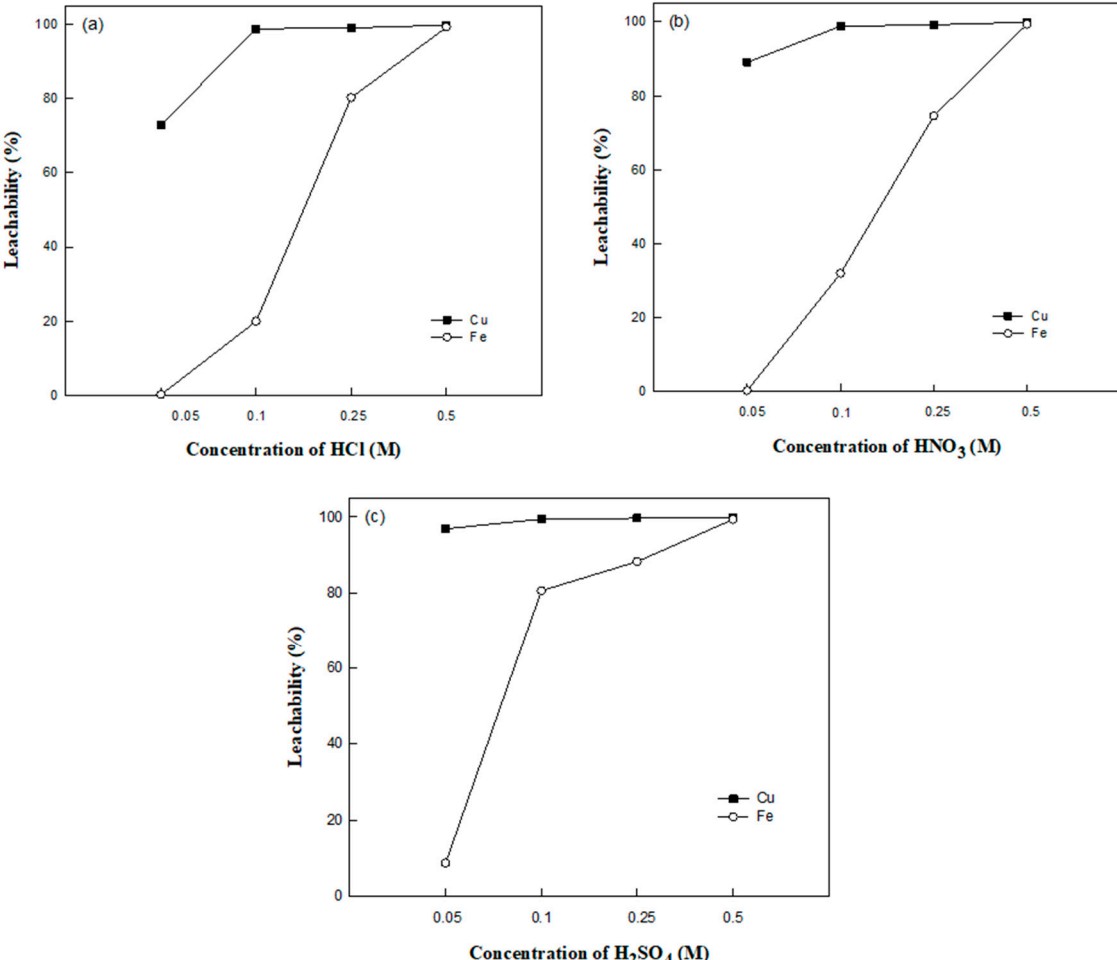

**Figure 4.** Cu and Fe leachability with variation of (**a**) HCl concentration, (**b**) HNO$_3$ concentration, (**c**) H$_2$SO$_4$ concentration.

### 3.3. Sulfuric Leaching of Copper Sludge

#### 3.3.1. Effect of H$_2$SO$_4$ Concentration

The effect of H$_2$SO$_4$ concentration on selective Cu leaching was further studied in the concentration range from 0.1 to 0.4 M for 60 min at 25 °C. All other parameters were kept constant (shown in Figure 5). At lower H$_2$SO$_4$ concentrations of 0.1 to 0.12 M (equilibrium pH ~4.0), no extraction of Fe was observed and only ~50% Cu was dissolved. Changing the H$_2$SO$_4$ concentration from 0.12 to 0.2 M, corresponding to a decrease of final pH from 3.96 to 2.17, improved the Cu leachability significantly from ~50% to 95%, while the Fe leachability only slightly increased up to ~5%. The critically low Fe leachability at this pH range was explained by the presence of Fe(III) hydroxide as the most major phase of Fe in the sample, which possesses low solubility at this pH (K$_{sp}$ = 6 × 10$^{-38}$) [21,22]. Increasing the H$_2$SO$_4$ concentration from 0.2 M (equilibrium pH 2.21) to 0.4 M (equilibrium pH 1.23) did not improve Cu leachability; however, the higher amount of acid dissolved effectively more Fe from the sample (from ~5% to 80.4%). The above results clearly indicate that the best separation of Cu and Fe was obtained at H$_2$SO$_4$ concentration of ~0.2 M. Further studies were devoted to determining the Cu leaching mechanism, neglecting the Fe extraction.

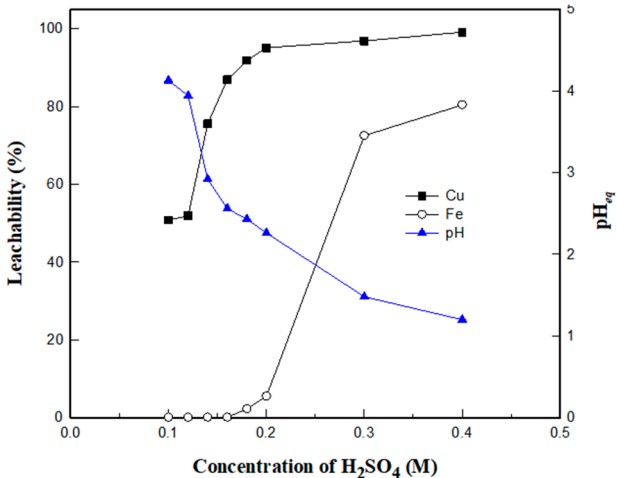

**Figure 5.** Effect of $H_2SO_4$ concentration on Cu and Fe leachability.

### 3.3.2. Effect of Leaching Temperature and Time

The effect of temperature on Cu leaching was investigated in the temperature range of 20 °C to 50 °C and a leaching time of 2 to 60 min, keeping other conditions constant. A 20-g sample in 500 mL of 0.18 M $H_2SO_4$ was used. The results plotted in Figure 6 show that Cu leaching increased with reaction temperature and time. Increasing the temperature from 20 °C to 50 °C enhanced the leachability from 5% up to 10%. The efficiency is already significantly high (77.2%) at the beginning of the leaching process and rapidly reaches ~85% after 10 min at 20 °C. The leaching improves only moderately with time, prolonging the reaction time from 10 to 60 min. The change of Cu leachability from 2 to 60 min was similarly observed in all temperature conditions used in this study, 20–50 °C. The leaching data, varying temperature and time, were further employed for kinetic and mechanistic studies based on shrinking core model assumptions.

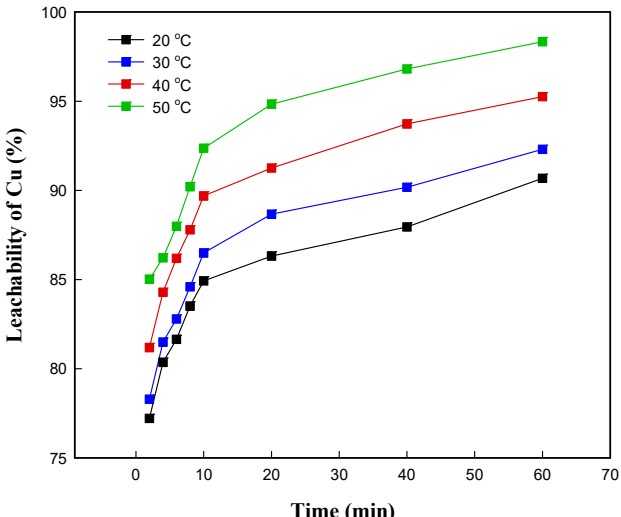

**Figure 6.** Effect of temperature and time on Cu leachability.

### 3.4. Kinetics and Mechanism of Copper Leaching in Sulfuric Acid

### 3.4.1. Kinetic Studies Based on the Shrinking Core Mode (SCM)

By assuming that the sample particle is spherical and has constant size during the reaction, the leaching of copper in $H_2SO_4$ solutions can be described as follows: (i) diffusion of $H_2SO_4$ from the bulk solution through the fluid film to the surface of the solid particle, (ii) diffusion of $H_2SO_4$ through the

solid product layer (the ash) to the surface of the unreacted core, (iii) reaction of $H_2SO_4$ with copper on the surface of the unreacted core, (iv) diffusion of leaching products through the ash layer back to the exterior surface of the solid particle, and (v) diffusion of products through the fluid film to the bulk solution [23]. In the case of shrinking spherical particles, the ash layer is not generated, and the solid particle size gradually decreases with time, meaning there is no diffusion of products through the ash layer before and after leaching. Based on these assumptions, the mechanism of leaching can be estimated following the kinetic equations below.

$$x = k_d \times t \quad \text{(Film diffusion control, constant particle size)} \tag{4}$$

$$1 - (1 - x)^{\frac{2}{3}} = k_d \times t \quad \text{(Film diffusion control, shrinking particle)} \tag{5}$$

$$1 - 3(1 - x)^{\frac{2}{3}} + 2(1 - x) = k_d \times t \quad \text{(Ash diffusion control)} \tag{6}$$

$$1 - (1 - x)^{\frac{1}{3}} = k_c \times t \quad \text{(Chemical control)} \tag{7}$$

where $x$ is the leaching efficiency at certain time $t$ (min), while $k_d$ and $k_c$ are the apparent rate constants (min$^{-1}$) for diffusion and chemical control, respectively.

The obtained rate constant is subsequently used to calculate the activation energy following the Arrhenius equation [24]:

$$k = A \times e^{\frac{-E_a}{RT}} \tag{8}$$

where $k$ is the rate constant obtained from the kinetic model equation, $A$ is a constant factor, $E_a$ is the apparent activation energy of leaching, $R$ is the universal gas constant (8.314 kJ/mol), and $T$ is the leaching temperature (K).

### 3.4.2. Determination of the Kinetic Model

The leaching data from 2 to 60 min were used in each model equation, and the fitted data obtained from Cu leaching did not perform the good linear relationship regarding Equations (4) to (7), showing a low $R^2$ regression coefficient of <0.8. For instance, the plot of $[1 - 3(1 - x)^{2/3} + 2(1 - x)]$ versus time (Figure 7) indicates that each curve at different temperatures can be divided into two lines, presenting the two stages of the leaching process. This suggested to separate the process into two stages, (2–10 min) and (10–60 min), respectively. The data are listed in Table 2. The film diffusion model with constant particle following Equation (4), did not match well both stages, while the film diffusion model with shrinking particle Equation (5) showed good regression coefficients ($R^2$), but the $R^2$ values for the calculated activation energy ($Ea$) were lower than those of the other two models Equations (6) and (7). The fitted data of Equations (6) and (7) show better regression coefficients ($R^2 > 0.97$); however, the best match was obtained with the ash diffusion model Equation (6). This model was most suitable to interpret the Cu leaching mechanism since the leaching efficiency was only slightly dependent on the variation of temperature and the activation energy ($Ea$) values were lower than 40 kJ/mol for all the leaching stages [24].

As a result, the fitted data of the ash diffusion model Equation (6) versus time were separated into two different stages (Figures 8 and 9), proposing a mechanism including two leaching platforms. The first stage was defined from the start till 10 min, where rapidly ~85% Cu leaching was obtained. The reaction rate constant varied from 0.0139 to 0.0201 min$^{-1}$ at 20 to 50 °C, respectively (Table 3). The second stage lasted from 10 to 60 min, with a lower rate reaction constant compared to the previous stage (Table 3), corresponding to a more gradual increase in Cu leaching efficiency. The relationship between ln$k$ and 1/$T$ is presented in Figure 10, and the obtained activation energy values were 9.8 kJ/mol for the stage below 10 min, and 20.9 kJ/mol for the subsequent stage from 10 to 60 min. It can be concluded that the leaching process within 10 min follows the mechanism of ash diffusion control, based on the model equation Equation (6), with an activation energy of less than 12 kJ/mol. The leaching stage between 10 min to 60 min was controlled by an intermediate model mechanism because the activation energy ranged between 12 and 40 kJ/mol [24].

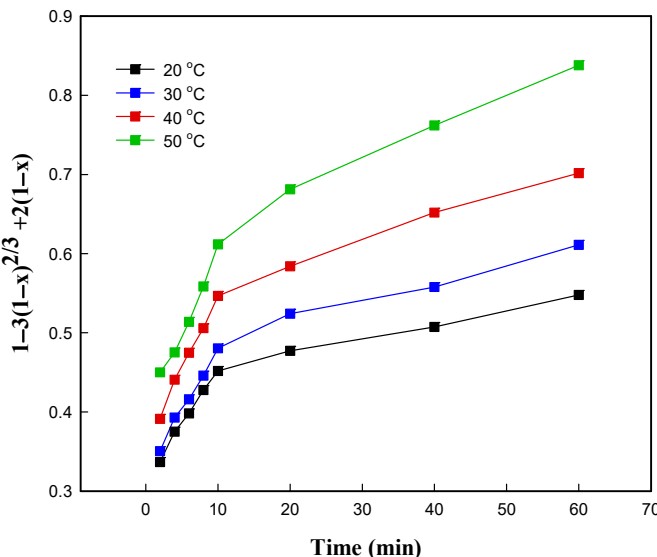

**Figure 7.** Plot of $[1 - 3(1-x)^{\frac{2}{3}} + 2(1-x)]$ versus leaching time at different temperature.

**Table 2.** Fitting leaching data to kinetic model equations in two leaching stages.

| Kinetic Model | Equation | Fitting Results | |
|---|---|---|---|
| | | Average Value of Regression Coefficient $R^2$ | Active Energy $E_a$(kJ/mol) and Regression Coefficient $R^2$ |
| **Stage 1: 2–10 min** | | | |
| Film diffusion control | $x = k_d \times t$ | Not fit | Not fit |
| Ash diffusion control | $1 - 3(1-x)^{\frac{2}{3}} + 2(1-x) = k_d \times t$ | 0.98 | 9.8 kJ/mol / $R^2 = 0.99$ |
| Chemical control | $1 - (1-x)^{\frac{1}{3}} = k_c \times t$ | 0.98 | 9.7 kJ/mol / $R^2 = 0.97$ |
| Film diffusion control | $1 - (1-x)^{\frac{2}{3}} = k_d \times t$ | 0.97 | 5.2 kJ/mol / $R^2 = 0.87$ |
| **Stage 2: 10–60 min** | | | |
| Film diffusion control | $x = k_d \times t$ | Not fit | Not fit |
| Ash diffusion control | $1 - 3(1-x)^{\frac{2}{3}} + 2(1-x) = k_d \times t$ | 0.98 | 20.9 kJ/mol / $R^2 = 0.99$ |
| Chemical control | $1 - (1-x)^{\frac{1}{3}} = k_c \times t$ | 0.97 | 25.47 kJ/mol / $R^2 = 0.97$ |
| Film diffusion control | $1 - (1-x)^{\frac{2}{3}} = k_d \times t$ | 0.96 | 19.6 kJ/mol / $R^2 = 0.93$ |

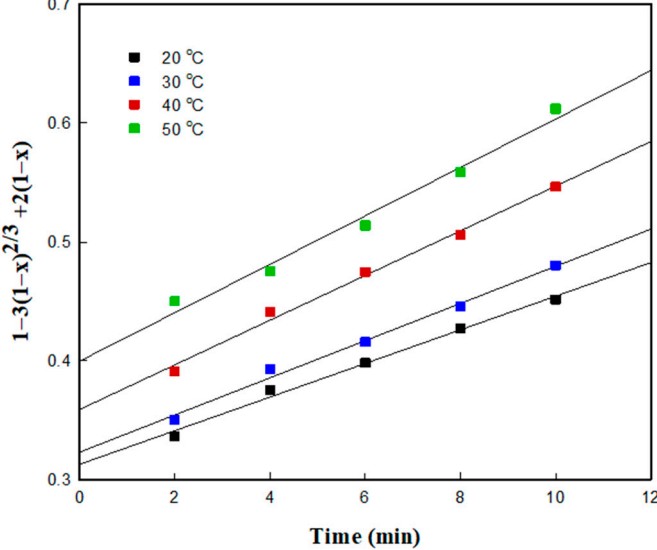

**Figure 8.** Plot of $[1 - 3(1-x)^{\frac{2}{3}} + 2(1-x)]$ versus leaching time (2–10 min) at different temperatures.

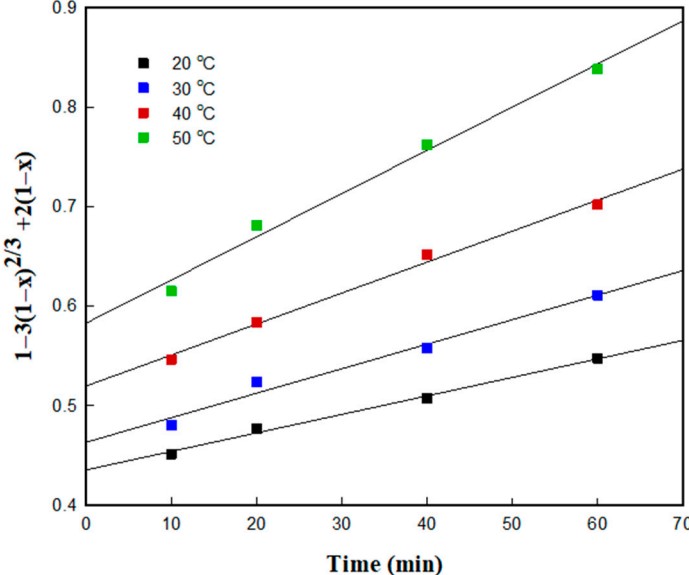

**Figure 9.** Plot of $\left[1-3(1-x)^{\frac{2}{3}}+2(1-x)\right]$ versus leaching time (10–60 min) at different temperatures.

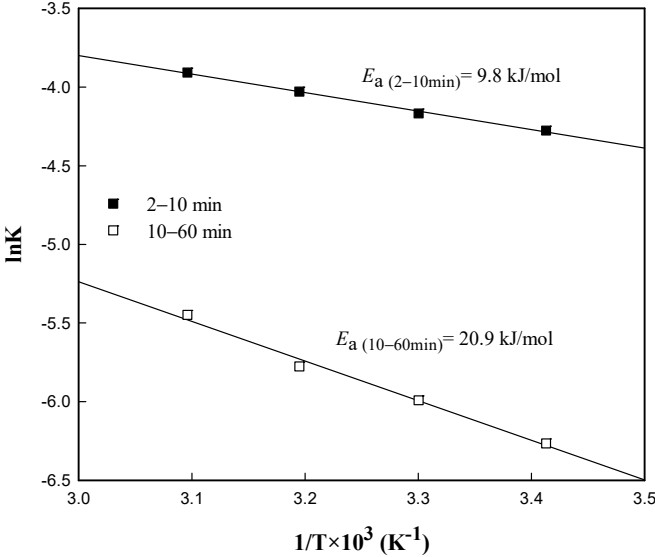

**Figure 10.** Plot of lnK versus 1/T to obtain the apparent activation energy.

**Table 3.** Value of apparent rate constant for ash diffusion control model in two leaching stages.

| Temperature (°C) | Apparent Rate Constant (min$^{-1}$) |
|---|---|
| **Stage 1: 2–10 min** | |
| 20 | 0.0139 |
| 30 | 0.0155 |
| 40 | 0.0188 |
| 50 | 0.0201 |
| **Stage 2: 10–60 min** | |
| 20 | 0.0019 |
| 30 | 0.0025 |
| 40 | 0.0031 |
| 50 | 0.0043 |

### 3.4.3. Mechanism of Cu Leaching from Waste Sludge in Sulfuric Acid

The interpretation of the Cu leaching mechanism based on the properties of the original sludge sample and the fitted kinetic model was proposed as below.

The leaching in Stage 1 is controlled by the diffusion of sulfuric acid through the porous layer on the surface of the sample particle, which is demonstrated in the SEM image of Figure 3. Sulfuric acid reacts with not only copper hydroxide but also other metal hydroxides, such as $Fe(OH)_3$, $Fe(OH)_2$, and $Ca(OH)_2$ (Equations (9)–(12)). The reactions between sulfuric and the metal hydroxide have a significantly high rate of reaction which supports the selection of the diffusion control model to interpret the leaching mechanism. The high leaching efficiency in this stage indicates the major presence of Cu in the hydroxide form because it is difficult to dissolve Cu metal with sulfuric acid only, without adding any oxidizing agents (Equation (13)).

$$Cu(OH)_2 + H_2SO_4 \rightarrow CuSO_4 + 2H_2O \quad \Delta G^0_{298} = -17.6 \, kcal/mol \tag{9}$$

$$Fe(OH)_2 + H_2SO_4 \rightarrow FeSO_4 + 2H_2O \quad \Delta G^0_{298} = -28.0 \, kcal/mol \tag{10}$$

$$2\,Fe(OH)_3 + 3\,H_2SO_4 \rightarrow Fe_2(SO_4)_3 + 6\,H_2O \quad \Delta G^0_{298} = -49.3 \, kcal/mol \tag{11}$$

$$Ca(OH)_2 + H_2SO_4 \rightarrow CaSO_4 + 2\,H_2O \quad \Delta G^0_{298} = -50.5 \, kcal/mol \tag{12}$$

$$Cu + H_2SO_4 \rightarrow CuSO_4 + H2 \quad \Delta G^0_{298} = 6.6 \, kcal/mol \tag{13}$$

Therefore, it can be concluded that the activation energy (9.8 kJ/mol) of the first leaching stage described the Cu dissolution in hydroxide form with sulfuric acid, following the ash diffusion model. This model was reported previously for the sulfuric acid-assisted Cu leaching from oxide ores containing the $Cu_2(OH)_2 \cdot CO_3$ phase [25,26]. However, the obtained activation energy was higher than that of the present work, 20.6 kJ/mol [25], and 26.69 kJ/mol [26]. The difference was attributed to the higher content of Ca (16.7%) and $SiO_2$ (37.04 to 69.2%) in the copper ores, compared to that of the copper sludge (Ca 6.24%, $SiO_2$ 0%), the latter being able to precipitate on the surface of the sample particle during leaching. It was reported that sludge containing copper hydroxide showed lower activation energy of 2.34 kJ/mol [27]; however, there was no mention of the influence of Fe compounds, because high Fe concentration can affect the penetration of sulfuric acid into the sample particle surface. Therefore, the properties of the original sample were the major reasons for activation energy variations in similar kinetic Cu leaching models.

In Stage 2, a critical amount of sulfuric acid was already consumed in the previous stage, and reducing the acidity from initially pH 0.56 to 2.24 (after 10 min), and 2.43 (after 60 min) resulted in the precipitation of iron hydroxides from ferric ion, following Equation (14). Ferric ion can dissolve copper metal following Equation (15) [28], having a smaller free Gibbs energy and lower reaction rate in comparison to Equation (9). As a result, the appearance of less soluble products (calcium sulfate and ferric hydroxide) could affect the sulfuric acid diffusion and the Cu leaching due to agglomeration on the sample particle surface. Consequently, Cu in metallic form was continuously leached into the solution at this stage, and this process had a lower reaction rate (Table 3) and a higher activation energy than in the previous stage (20.9 kJ/mol). The high activation energy was actually close to that obtained for Cu metal leaching from PCBs in acid solution, 20.7 kJ/mol [29], or that in the presence of ferric sulfate, 18.3 kJ/mol [28].

$$Fe^{3+} + 3OH^- \rightarrow Fe(OH)_3 \quad K_{sp} = 6 \times 10^{-38} \tag{14}$$

$$Fe_2(SO_4)_3 + Cu \rightarrow 2FeSO_4 + CuSO_4 \quad \Delta G^0_{298} = -11.4 \, kcal/mol \tag{15}$$

## 4. Conclusions

The selective recovery of copper from copper-containing sludge was investigated, using a simple acid leaching method. The sample, generated from PCBs manufacturing process and composed mainly

of Cu and Fe in the form of $Cu(OH)_2$ and $Fe(OH)_3$, was characterized using XRD, thermogravimetry, and differential thermal analysis. Varying the acid (HCl, $HNO_3$, $H_2SO_4$) concentration from 0.05 M to 0.5 M revealed a high potential for separating Cu from Fe at lower acid concentrations. Cu leaching increased in the order of HCl < $HNO_3$ < $H_2SO_4$. The highest selectivity for Cu leaching (~95%) and the lowest Fe dissolution (~5%) was observed, using ~0.2 M $H_2SO_4$ at 25 °C for 60 min. The kinetics proposes a two-stage leaching mechanism at (2–10 min) and (10–60 min), corresponding to the leaching of Cu as $Cu(OH)_2$ and Cu metal, respectively, additionally influenced by Fe in the form of $Fe(OH)_3$. The first stage has lower activation energy (9.8 kJ/mol) than the second one (20.6 kJ/mol), which indicated that Cu leaching followed the ash diffusion and the mixed control model, respectively. The present work provided a simple process to recover selectively copper from copper-containing sludge, reducing chemical consumption and separation stages. This recycling process is of importance to the metallurgical industry because it uses secondary resources to sustain further the rapid development of our high-technology society.

**Author Contributions:** Conceptualization, J.L.; methodology, J.L. and H.B.T.; formal analysis, S.K. and H.B.T.; investigation, S.K. and H.B.T.; writing—original draft preparation, H.B.T.; writing—review and editing, J.L.; visualization, J.L.; supervision, J.L. All authors have read and agreed to the published version of the manuscript.

**Funding:** This study was supported by the R&D Center for Valuable Recycling (Global-Top R&BD Program) of the Ministry of Environment. (Project No.:2019002220002).

**Acknowledgments:** The authors would like to thank Jonghyun Kim (UST-KIGAM) for his support in the SEM-EDS analysis of the samples.

**Conflicts of Interest:** The authors declare no conflict of interest. The funders had no role in the design of the study; in the collection, analyses, or interpretation of data; in the writing of the manuscript, or in the decision to publish the results.

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
