# Peer review of "Selective Copper Recovery by Acid Leaching from Printed Circuit Board Waste Sludge"

_metals, doi:10.3390/met10020293_

Round 1
Reviewer 1 Report
The authors described their experimental activities in order to selectively dissolve copper from the sludge of printed circuit boards by different acid leaching methods. The aim was to obtain high copper extraction yields without iron dissolution. Finally, kinetics studies were carried out on the mechanism of copper leaching by sulfuric acid (the more suitable acid as confirmed by experimental results).
The research is planned appropriately and, in my opinion, is of good interest to the readers of the journal. Anyway, some minor improvements are necessary.
The introduction can be improved, especially would be necessary adding more recent papers. Although, several characterization analyses have been already conducted and the section “Materials and Methods” is well written, I suggest adding more information regarding the processes which the sludge of PCBs was previously subjected. In addition, after the grinding by rod mill, authors state that the fraction of particle size of 45 µm was collected; is there also a fraction above to 45 µm? Has it been physically and chemically characterized? Some experimental data showed in figures 4 C, 5 and 6 seem not comparable. For instance, in test Figure 4 C, a dissolution of 100 % of copper is obtained at 60 min, 25 °C and 0.1 M of sulfuric acid, while in Figure 6 at 0.18 M of sulfuric acid, 60 min and 20 °C a copper dissolution of about 80 % is achieved. It depends by the change of pulp density or by the test replicability? What is author explanation? In order to increase the value of this research at industrial scale, authors think that similar results can also be obtained selecting a higher pulp density than those investigated?
Author Response
We are grateful to the reviewers for kindly offering invaluable comments. According to the comments and instruction from the editor, we have revised the paper as follows:
Comment 1: The introduction can be improved, especially would be necessary adding more recent papers.
Answer) The recent papers have been added to improve the introduction (Line 50-56; line 68-72).
Comment 2: Adding of the information regarding the processes which the sludge of PCBs was previously subjected.
Answer) The sample was collated from the wastewater treatment factory and provided by the company SungEel Himetal, South Korea without any further treatment. Therefore, the information regarding generation processing of the sample was not supplied. However, the description has been added in the “Materials and methods” section based on the common process to produce the heavy metal-containing sludge during PCBs manufacturing from researches of literature review (line 87-90).
Comment 3: Characterization of sample with fraction above 45μm
Answer) The characterization of sample with particle size above 45μm has been analyzed with repetition of 3 times, however, this results is much lower than that of sample with fraction -45μm and not consistency which indicates the complexities of sample before screening and it is not reasonable for further investigation (given in below table). Furthermore, the sample was obtained from the sedimentation using chemical processing which certainly produced the very fine particle of heavy metal hydroxides, and the larger size compositions were other impurities of fibers or organic compounds. The impurities should be separated from the metal-containing particles by sieving before leaching. Therefore, the investigation of particle size above 45μm was excluded.
|
Time |
Metal concentration (%) |
|
|
Fe |
Cu |
|
|
1 |
14.9 |
5.6 |
|
2 |
7.3 |
4.5 |
|
3 |
10.2 |
10.9 |
Comment 4: Experimental replicability
Answer) The experiments to investigate the replicability of Cu and Fe dissolution have been conducted at same conditions of leaching, and it show good consistency of efficiency value for both Cu and Fe with the standard deviation range from only ±1.4% to ±3.8%, which was not presented in the manuscripts for the sake of brevity. Therefore, the difference of Cu leaching from Figure 4C and 6 related to the variation of pulp density.
Comment 5: Selecting of higher pulp density
Answer) In order to increase the value of this research at industrial scale, higher pulp density should be concerned, however, the higher pulp density leads to larger amount of residue which result more difficulty in filtration stage after leaching. The purpose of this study is separation of Cu from Fe by leaching Cu in leach liquor and remaining Fe in the residue, hence the filtration stage should be concern as well. Therefore, the pulp density was not investigated at higher value. We believe that the manuscript being a basic study, has achieved the objectives intended to selectively extract Cu from the sludge sample with description of leaching mechanism. Though it is now certainly planned and conducted in near future to develop at industrial scale.
Reviewer 2 Report
High copper extraction efficiency and low iron selectivity were obtained in one leaching step. This was achieved without the use of any other oxidizing or precipitating agents, which is the most significant advantage of the presented work. In addition, kinetic studies were described to investigate the mechanism of copper dissolution under selected leaching conditions.
The change in the concentration of acids used in the study (HCl, HNO3, H2SO4) from 0.05 M to 0.5 M showed great potential for separating Cu from Fe at lower acid concentrations. Increased Cu elution occurred with H2SO4. The highest selectivity of Cu leaching at the level of 95% and the lowest dissolution of Fe ~ 5% were observed, using ~ 0.2 M H2SO4 at 25°C for 60 minutes.
The multiplication in mathematics is marked "·- the work has the characters "*" "."; suggest to harmonize the designation.
Author Response
We are grateful to the reviewers for kindly offering invaluable comments. According to the comments and instruction from the editor, we have revised the paper as follows:
comment) the expression of multiplication in equation
Answer) we corrected our manuscript as you suggested.
Reviewer 3 Report
Dear authors,
The manuscript “Selective copper recovery by acid leaching from printed circuit board waste sludge” Trinh et al., investigates the leachability and of the copper in a recycling product sludge under various conditions aiming, at the same time, to the limitation of iron leaching. The subject of the current work is interesting and it accompanied by a kinetic study. However, some weak points should be stressed out and some additional information are necessary in order the article improved prior to its publication:
- Information concerning the sludge should be quoted. How it is generated? Through which process? Does it consists a by-product of a recycling process?
- Why do you choose the fraction -45 μm? Does the chemical analysis of table 1 referred to this fraction? The chemical analysis of +45 μm should also be given.
- X ray diffractometry and SEM gives poor data concerning the copper and iron rich compounds. It is necessary to repeat SEM-EDS analysis using polished powder samples in order particles of different phases be distinguished.
- According to the kinetic study, why the “chemical control” model is excluded? Ea presents similar correlation coefficient values in both “chemical control” and “ash diffusion control” models. This it should appropriately be documented.
Author Response
We are grateful to the reviewers for kindly offering invaluable comments. According to the comments and instruction from the editor, we have revised the paper as follows:
Comment 1: Information concerning the sludge should be quoted. How is it generated? Through which process? Dose it consist a by- product of a recycling process?
Answer) The sample was collated from the wastewater treatment factory and provided by the company SungEel Himetal, South Korea without any further treatment. Therefore, the information regarding generation processing of the sample was not supplied. However, the description has been added in the “Materials and methods” section based on the common process to produce the heavy metal-containing sludge during PCBs manufacturing from researches of literature review(line 87-90).
Comment 2: Why do you choose the fraction -45μm? Dose the chemical analysis of table 1 referred to this fraction? The chemical analysis of +45μm should be given.
Answer)The characterization of sample with particle size above 45μm has been analyzed with repetition of 3 times, however, this results is much lower than that of sample with fraction -45μm and not consistency which indicates the complexities of sample before screening and it is not reasonable for further investigation (given in below table). Furthermore, the sample was obtained from the sedimentation using chemical processing which certainly produced the very fine particle of heavy metal hydroxides, and the larger size compositions were other impurities of fibers or organic compounds. The impurities should be separated from the metal-containing particles by sieving before leaching. Therefore, the investigation of particle size above 45μm was excluded.
|
Time |
Metal concentration (%) |
|
|
Fe |
Cu |
|
|
1 |
14.9 |
5.6 |
|
2 |
7.3 |
4.5 |
|
3 |
10.2 |
10.9 |
Comment 3: XRD and SEM give poor data concerning the copper and iron rich compounds. It is necessary to repeat SEM-EDS analysis using polished powder sample in order particles of different phase should be distinguished.
Answer) We do agree with the reviewer about doing SEM-EDS of polished powder to distinguish the particle phases. However, the sample was too light and already collected at very fine particle size, which is not easy to process the polishing stage and also difficult to make the sample by molding technique. There were many attempts to improve the analysis, but the results of SEM-EDS were low quality which cannot be presented to support the SEM-EDS of original sample. Though it is not possible at this stage, we do believe that the characterization of sample is possible to confirm the phase of metals which is fundamental to demonstrate the separation possibility of Cu from Fe and leaching mechanism. We appreciate for the suggestion to improve the manuscript and would like the reviewer to consider the point of benefit to the whole process; and will be in agreement for the publication.
Comment 4: According to the kinetic study, why the “chemical control” model is excluded? Ea presents similar correlation coefficient values in both “chemical control” and “ash diffusion” control models. This is should appropriately be documented.
Answer) Although correlation coefficient values in both “chemical control” and “ash diffusion” control models, the dependency of Cu leaching efficiency on temperature show the manner of ash diffusion control models. A diffusion control model is slightly dependent on temperature whereas the chemical control model is significantly dependent on temperature. It is proved by the enhancement of Cu leaching was not remarkable when increasing of temperature from 20 oC to 50 oC. Therefore, the value of activation energy of diffusion control model is moderate 4 to 12 kJ/mol while the chemical control has much higher value > 40 kJ/mol. The ash diffusion control model proposed for mechanism of Cu leaching was also based on the nature of sample contained several metal hydroxides which can dissolved with H2SO4 with high value of rate reactions (given as below). The diffusion control has been not only first time selected to interpret the leaching mechanism from Cu sludge, but also in other previous investigations which were included in the manuscript. Therefore, the chemical control model was not suggested to be used in leaching mechanism description.
Cu(OH)2 + H2SO4® CuSO4 +2H2O logK= 12.9
Fe(OH)2 + H2SO4® FeSO4 +2H2O logK= 20.5
2 Fe(OH)3 + 3 H2SO4 ® Fe2(SO4)3 +6 H2O logK= 36.1
Ca(OH)2 + H2SO4® CaSO4 +2 H2O logK= 37.0
The addition was already included in line 363-364, and line 493-495.